# Consumer Choices and Habits Related to Tea Consumption by Poles

**DOI:** 10.3390/foods11182873

**Published:** 2022-09-16

**Authors:** Ewa Czarniecka-Skubina, Renata Korzeniowska-Ginter, Marlena Pielak, Piotr Sałek, Tomasz Owczarek, Agata Kozak

**Affiliations:** 1Department of Food Gastronomy and Food Hygiene, Institute of Human Nutrition Sciences, Warsaw University of Life Sciences (WULS), Str. Nowoursynowska 166, 02-787 Warsaw, Poland; 2Department of Quality Management, Gdynia Maritime University, Str. Morska 81-87, 81-225 Gdynia, Poland; 3Department of Management and Economics, Gdynia Maritime University, Str. Morska 81-87, 81-225 Gdynia, Poland

**Keywords:** consumer habits, tea, tea brewing, consumer preferences, Poles, CAWI method

## Abstract

Tea is one of the most consumed beverages in the world. In the literature, much attention is paid to the influence of tea and its components on human health and consumer purchasing behavior. The aim of the study was to analyze the habits of Polish consumers regarding tea consumption, brewing methods, and their choices related to tea, to describe the characteristics of tea consumers, and present their segmentation based on consumer choices and habits regarding tea consumption. The study was performed using the computer-assisted web interviewing (CAWI) method on a group of 1700 adult consumers of tea. Information about consumer choices and habits related to tea consumption was collected, including brewing method, place of tea consumption, and factors determining tea choices. Using cluster analysis, six groups of tea consumers were identified. These are “Occasional tea gourmets”, “Yerba mate drinkers”, “Tea gourmets”, “Occasional consumers”, “Undemanding tea consumers,” and “Occasional strong tea consumers”. In summary, it can be said that Poles are not tea gourmets; they prefer black tea, in bags, brewed in cups or glasses for up to 3 min, and usually drink teas without any additives, at home, several times a week, during breakfast and between meals. The most popular brand among the respondents was Lipton. Consumers have little knowledge of the health benefits of tea.

## 1. Introduction

Tea is one of the most consumed beverages in the world for its health, sensory, stimulant, relaxing, and cultural properties. Depending on the place in the world, the preferences related to the method of brewing and serving tea are different [1].

The tea culture was established in China. During the late eighth century, it was introduced in Japan and became an important part of Japanese culture. Tea is also recognized as one of the cultural beverages in the United Kingdom, also in Middle Eastern and North African cultures, as well as in Russia [1,2] and even in Germany, where the East Frisian tea culture has the uniqueness of being the only UNESCO-awarded tea consumption tradition worldwide, in 2016 [3]. Incomes and price have been important but not always paramount factors in determining whether tea or coffee is predominant. Tea was the preferred drink in Great Britain, Australia, New Zealand, and Canada for a long period when they were among the richest countries in the world [1].

Most of the research in the scientific literature has focused on the potential health benefits of tea and its ingredients such as polyphenols, and catechins, especially for low-processed teas such as, white, green, and red tea [4,5,6,7,8,9]. Due to its numerous health properties, i.e., antioxidant, thermogenic, anti-inflammatory, cholesterol-lowering, antimicrobial, neuroprotective, antihypertensive, and anti-carcinogenic, the presence of tea in the human daily diet is significantly high [10,11,12,13,14,15,16,17,18,19,20,21,22]. The health benefits of tea consumption in humans have been documented extensively, including preventing cancer [10,11], cardiovascular diseases [9,10,12,13], coronary disease [23], Alzheimer’s disease [24,25,26], depression [27,28,29,30], and lowering COVID-19 morbidity/mortality [31,32,33,34]. Studies have also shown that tea consumption reduces the risk of getting flu and upper respiratory infections [35], aids weight loss [4,36,37], has a marked effect on bones as well as being a good source of vitamin K [38,39,40]. Drinking too much tea can raise blood pressure and cause kidney stones due to the high oxalate content which increases with long brewing times and strong infusions [41,42]. Yerba Mate is not a tea and is obtained from the evergreen holly (*Ilex paraguariensis* A. St.-Hil.), but the growing interest of consumers in antioxidant-rich infusions can be explained by the common fashion for a healthy lifestyle, as shown by both commercial data and studies [43,44]. China was the leading global tea producer, followed by India and Kenya. Worldwide tea production amounted to around 5.2 million metric tons in 2015 and 6.5 in 2019 [2]. Black tea is consumed mainly in Europe, North America, and North Africa (except Morocco), while green tea is drunk across the whole of Asia, and Oolong tea is also popular in China and Taiwan. World tea consumption has increased by 3.5% over the last decade and is projected to increase further by 2% over the next decade [45]. Black tea is the most important type of tea in international trade [46].

Tea is one of the most popular beverages consumed all over the world [47], including in European countries [1]. Annual tea consumption in Turkey is the largest in Europe and the world and is stated to be 3.16 kg per capita. Ireland ranks second in terms of tea consumption in Europe. The British, despite having a reputation for being big tea drinkers, are only the 3rd biggest tea drinkers in Europe. Southern European countries (except Turkey) consume the lowest amount of tea per capita. Average annual tea consumption in Europe ranged from 0.76 kg per capita (in 2012) to 0.80 kg (in 2021) at home, and 0.04 kg per capita outside of the home, only in 2017–2019 was it 0.05 kg per capita. The average volume per person of tea in Europe in 2022 is estimated to amount to 0.72 kg per capita. It is expected that by 2025, 51% of spending and 6% of tea consumption will be attributable to out-of-home consumption (e.g., in bars and restaurants) [48,49].

Tea consumption in Polish households ranges from 0.84 kg per capita in 2008–2011, up to 0.72 kg per capita in 2012–2015, and 0.60 kg per capita in 2016–2020 [50,51,52,53,54,55,56,57,58,59,60,61,62]. It is expected that by 2025, 39% of spending on tea and 3% of tea consumption will be attributable to out-of-home consumption (e.g., in bars and restaurants). It is estimated that in 2022 the average consumption of eta per capita in Poland will amount to 0.51 kg [63].

About 80% of global tea consumption is made up of the black variety, the remaining 20% of green, oolong, red, and yellow [64]. In the Polish market, the leader is black tea, which accounts for approximately 70% of the sales volume, with a 7% share of green tea and a 1.5% share of red tea [43].

The habits and behavior of the modern consumer in the tea market depend, among others, on psychological criteria (motivation and needs as well as pleasure of consumption), behavior (culture of life, frequency, and number of cups consumed, method of its preparation and place of consumption), descriptive (gender, age, professional activity, family size, dwelling place, and economic situation) and marketing (advertising and prices) [65]. Tea is seen as an essential part of the daily diet, and its consumption has been recognized as a “pro-health habit” [66].

In the scientific literature, however, there is little [67,68,69,70,71,72,73,74] consumer research on the habits and preferences of tea consumers, the way of brewing, and the additives to tea used by consumers. It should be emphasized that contemporary customers have more knowledge in the field of nutrition and their behaviors change with the increase in this knowledge. Therefore, our research aimed to analyze Polish consumers’ habits towards tea consumption, their choices and preferences related to tea, and the way of brewing methods. The second goal of this study is to identify, describe, and compare consumer segments based on differences in individual tea choices and tea habits.

## 2. Materials and Methods

### 2.1. Data Collection

The computer-assisted web-based interviewing (CAWI) method was used to collect all data. The survey was conducted on a group of 1700 adults in Poland, declaring their consumption of tea. The sample is representative of all adult residents of Poland. The respondents completed the questionnaire online. A link to the questionnaire in Polish language Google Forms format was sent via Facebook, WhatsApp^®^, e-mail and students forum. A questionnaire provided on a webpage increases the sense of anonymity and gives an opportunity to participate in the study at a time convenient for the respondent. The questionnaire was designed based on previous research related to tea consumption [67,68,69,70,71].

The questionnaire was validated by means of a pilot study with 20 people. Any problems were identified, and the questionnaire was completed and amended. It was estimated in a pilot test that completing the form would take each participant around 15 min. Each adult respondent who agreed to take part in the study was invited to fill in the questionnaire. The respondents were free to participate in the research. Because the research was non-invasive and details of the participants remained undisclosed, the research does not fall within the remit of the Helsinki Declaration.

### 2.2. Questionnaire

The questionnaire consisted of two parts of which the first part had nineteen questions relating to tea consumption and consumer habits with tea. The questions concerned the preferences and frequency of consumption of various types of tea, factors determining the choice and purchase of the tea brand, tea brewing (amount of tea, type of water, brewing method, and time), the use of sugar, sweeteners, and others, as well as knowledge about the health effects of tea, and the places and occasions of its consumption. The second part of the questionnaire consisted of five questions that were related to a respondent’s sociodemographic details (gender, age, education, dwelling place, and financial status).

### 2.3. Characteristics of Respondents

Characteristics of respondents are presented in Table 1.

The study mainly involved women, with secondary or high education, and different dwelling places. The financial status of respondents was average or good (85.41%). Respondents were in the age range between 18–75 years old, had access to a computer, the internet, and had sufficient computer literacy skills.

### 2.4. Data Analysis

The statistical analysis of the results was performed using Statistica software (StatSoft Inc., Hamburg, Germany, version 13.3 PL).

The study primarily used cluster analysis to isolate groups of respondents with similar attitudes toward tea purchase and consumption. In the first stage, the original number of nearly 100 variables was reduced to a few or a dozen new variables. This action was carried out to make the results more transparent and easier to interpret. The goal was to reduce the number of variables without significant loss of information carried by them. Due to the ordinal nature of most of the variables, an agglomeration method was used to determine the number of new variables, with complete linkage (distance of the furthest elements) and percentage discrepancy as a measure of distance. Homogeneous groups of variables were extracted using the *k*-means method. Further analysis used new variables created as averages of variables in groups calculated on a case-by-case basis. In the second part of the analysis, clusters of respondents were created due to similar values of the new variables. In this case, too, the agglomeration method and the k-means method were used. The groups thus formed were characterized using descriptive statistics, in particular medians (Me) and interquartile ranges (IQR). For a more precise interpretation of the results, arithmetic means were also used. They were used only to order the results obtained.

Non-parametric tests of significance for differences in average values (medians) across groups were used to determine the significance of the effect of demographic factors on tea consumption behavior. The Mann–Whitney U test was used for two median values, while the Kruskal–Wallis H test (also called K–W ANOVA) with post-hoc analysis was used for multiple groups. All tests were performed with a significance of *p* < 0.05.

## 3. Results

### 3.1. Type, Frequency, and Place of Tea Consumption

All the participants in the study (n = 1700) declared drinking tea, with not very high frequency (Table 2). The greatest number of people declared the consumption of black, green, and flavored tea. The respondents most often drank black tea, i.e., several times a week (median 5). Green tea, flavored tea, and beverages, that respondents defined as “fruit or herbal tea”, are drunk one to three times a month (median 3). Women consumed fruit or herbal infusions significantly more often (*p* = 0.005). Green tea is significantly more often consumed by women (*p* = 0.00001), people up to 40 years old (*p* = 0.000001), and living in large cities of over 100 thousand inhabitants (*p* = 0.0012). Yerba mate is also treated by the respondents as tea and is much more popular among people aged 25–40 years (*p* = 0.0007), living in large cities (*p* = 0.0031). White and red tea are significantly more often consumed by women aged 25–40, with higher education, resident in large cities, and with a very good financial situation (*p* < 0.05). Oolong tea is significantly more often drunk by people aged 25–60 years (*p* = 0.00002).

Among the respondents, the most popular is express tea (n = 1033, 60.8%), less popular is both whole-leaf tea (n = 303, 17.8%) and crushed leaf tea (n = 319, 18.8%), the least popular is granulated tea (n = 37, 2.2%) and others (n = 8, 0.5%) such as blooming tea balls. People over 60 years of age significantly more often prefer leaf tea (*p* = 0.00001), between the ages of 40–60 years crushed leaf tea (*p* = 0.0006), and women or people up to 40 years of age prefer express tea (*p* = 0.0001).

Tea, regardless of demographic factors, is usually consumed by respondents at home (n = 1385, 81.5%). They drink it less often at work (n = 212, 12.5%) or with friends and family (n = 88; 5.2%). It is not popular to drink it in a gastronomic establishment, i.e., a cafe, confectionery shop, tea house (n = 12, 0.7%) or in a canteen (3, 0.2%). At work, tea is most often drunk by people aged 41–60 years with higher education, while with friends and family, it is people over 61 years (p = 0.00001).

The respondents most often drank it between meals (n = 1144, 67.3%), during the first breakfast (n = 1057, 62.2%) or dinner (n = 826, 48.6%). Less often it is drunk during second breakfast (n = 393, 23.1%); lunch (n = 283, 16.7%) or during afternoon tea (n = 307, 18.1%).

### 3.2. Factors Determining the Choice of Tea

Among factors determining the choice of tea by respondents, the quality of tea (n = 1300, 76.5%) was mentioned. The price (n = 971, 57.1%), habits/preferences (n = 815, 47.9%), brand (n = 781, 45.9%), and sensory attributes (n = 658, 38.7%) were also quite important. Less important for the respondents were health reasons (n = 473, 27.8%), promotion (n = 377, 22.2%), as well as packaging (attractiveness, information, n = 327, 19.2%) or convenience (n = 259, 15.2%), and packaging size (n = 254, 14.9%). Factors of no significance in influencing the purchase of tea were: friends’ opinion (n = 85.5%), advertisements (n = 84, 4.9%), and the presentation on a shelf in a store (n = 85, 5%). Men significantly more often pay attention to tea brand, advertising, and convenience, while women to health reasons (*p* < 0.05). Price, packaging, promotion, and friends’ opinion were more significant for people under 24 years, with vocational or primary school education (*p* < 0.05).

The most important features of tea, regardless of demographic factors, were primarily the taste (n = 1614, 94.9%) and the aroma (n = 1219, 71.7%) of the tea. Color (n = 306, 18%) and other tea features (n = 36, 2.1%) were less important. Among other features, the respondents indicated the properties of tea such as astringency and strength of infusion.

The brand of tea is an individual preference of respondents. Tea selection depended on financial status (*p* < 0.05). More expensive brands, such as Lipton and Dilmah, were significantly more often chosen by people with a good or very good financial situation (*p* < 0.05), while cheaper brands such as Saga are chosen by people in a poor financial situation (*p* = 0.000001). The most popular tea brand chosen by respondents was Lipton (n = 1185, 69.7%). Less popular were: Dilmah (n = 663, 39%), Tetley (n = 415, 24.4%), Saga (Unilever, n = 330, 19.4%), Teekanne (n = 280, 16.5%), and Twining’s (n = 174, 10.2%). The respondents also choose tea blends (n = 104, 6.1%) from specialist tea stores and own brands of hyper- and supermarket chains operating in Poland (n = 91, 5.4%). Other brands were named by 294 respondents (n = 17.3%). Among them were Brooke Bond (n = 39, 2.3%), Posti (n = 35, 2.1%), Mokate (n = 25, 1.5%), and TET (n = 9, 0.5%).

### 3.3. The Methods of Preparing and the Kind of Tea Used by Respondents

The method of brewing is not important for the respondents (Table 3). Only 32.2% of respondents pay attention to it, and 37.6% only occasionally. When making tea, they use tap or filtered water. They drink weak tea, usually using 1 to 1.5 teaspoons of tea (67.5% of respondents) for brewing. The tea is usually brewed for up to 5 min, with a small percentage of respondents brewing for longer (Table 3). Men significantly more often use 3 g to brew tea, and women 2 g (*p* = 0.00001).

The respondents, regardless of the socio-demographic data (*p* > 0.05), brew tea mainly in a mug or in a cup after pouring boiling water (n = 1445; 85%). A small percentage of them consume iced tea (n = 189, 11.1%), and prepare green tea in a special teapot, pouring water over it several times (n = 182, 10.7%), or prepare an infusion and boiling water separately, usually in a teapot (n = 159, 9.4%). Other, national brewing methods are used by a smaller percentage of respondents. The respondents listed tea brewing in Bavarian style (n = 49, 2.9%), English (n = 39, 2.3%), Japanese and Moroccan (n = 35, 2.1%, respectively), Russian (n = 24, 1.4%), and even in Tibetan (n = 3, 0.2%). Other methods, such as cold brew or French press, were mentioned by 152 respondents (8.9%).

A significant percentage of the respondents (n = 999, 58.8%, *p* < 0.05) do not use sugar and sweeteners in tea. Only 701 people (41.2%) sweeten their tea with sugar or sweeteners such as xylitol, erytrol, or steviol glycosides (stevia). Sweeteners were used by 50 respondents (3%). One teaspoon of sugar is used by 403 respondents (23.7%) and two or more teaspoons of sugar by 248 respondents (14.5%). Significantly more often sugar is used to sweeten tea by men (*p* = 0.0001), the youngest (*p* = 0.0001), village residents (*p* = 0.0004) and people with primary or vocational education (*p* = 0.000001).

Other tea additives are used by 52.6% (n = 894) of respondents, mainly lemon (n = 807, 47.5%), honey (n = 522, 30.7%), and various fruit juices or syrups (n = 306, 18%) including raspberry, elderberry flower juice, elderberry juices, linden juices, quince syrups, or others. Milk for tea is used by only 70 respondents (4.1%) and alcohol by 42 respondents (2.5%). Other additives (n = 130, 7.7%) included spices: ginger, turmeric, cinnamon, cloves, cardamom, anise, herbs: mint, and fruits: orange, apples, and raspberries.

Women (*p* = 0.0098), people aged 25–60 years (*p* = 0.0007), with higher education (0.00001), inhabitants of large cities (*p* = 0.0004), with a very good financial situation (*p* = 0.00004) significantly more often do not use any additives to tea, enjoying its pure properties. Women significantly more often add lemon to tea (*p* = 0.0022) as well as fruit juice (*p* = 0.0329), while men with basic education more often add -alcohol (*p* = 0.00001). Honey is significantly more often added to tea by people aged up to 40 years (*p* = 0.00001) with secondary education (*p* = 0.0254).

### 3.4. Knowledge about Tea by Respondents

The respondents drink tea mainly for pleasure and sensory values, not for health reasons. Various diets ordered by a physician are used by a small percentage of the respondents (n = 59, 3.5%) and by their own choice, slimming, and vegetarian diets (n = 183, 10.8%).

They assess their nutritional knowledge as both good and very good (n = 1033; 60.8%) and insufficient or average (n = 667, 39.2%). However, the knowledge about the properties of tea among the respondents is not very high, as indicated by the obtained results (Table 4). They mainly assessed their diet as good or very good (n = 1408, 82.8%). A smaller percentage of respondents (n = 292, 17.2%) assessed it as insufficient or average.

### 3.5. Clusters and the Impact of Demographics on Consumer Behavior

Originally, there were 93 variables in the survey. Two variables were removed due to identical values (lack of variability), six more due to reference to “other” responses that cannot be interpreted quantitatively. The remaining 85 variables were used in the analysis. However, there are too many of them and they would have caused great difficulties in interpreting the results obtained. It was decided to create new artificial variables corresponding to groups of variables with similar responses. The optimal number of new variables was obtained using the agglomeration method. Its results can be found in Figure 1. Figure 1 shows the dendrogram of the agglomeration of variables, showing groups of variables more and less similar to each other. The lower the linkage between the vertical lines in the chart, the stronger the relationship between the variables. By appropriately selecting the level of dissimilarity, it is possible to indicate the optimal number of groups of variables. Assuming a dissimilarity of 0.7, it can be assumed that the optimal number of new variables is 10.

Agglomeration of variables using the *k*-means method resulted in 10 clusters of variables with similar values in the sample. For each cluster, arithmetic averages of the values for each observation were calculated. This resulted in 10 new synthetic variables (Cl01-Cl10). These variables are artificial creations and have no natural interpretation. However, it is possible to determine what the characteristics included in these variables refer to: Cl01—frequency of red tea consumption; Cl02—frequency of abnormal behavior toward tea; Cl03—frequency of consumption of yerba mate; Cl04—frequency of typical (for Poles) behaviors towards tea; Cl05—tea as a household daily beverage; Cl06—paying attention to things related to tea, diet and health; Cl07—frequency of consumption of white tea; Cl08—tea brewing time; Cl09—frequency of consumption of popular types of tea (black, green, fruit and herbal teas); Cl10—knowledge of the health properties of tea.

Using 10 new variables, a cluster analysis was conducted to select clusters of respondents giving similar answers. The analysis used the agglomeration method, taking percentage discrepancy and complete linkage clustering as a measure of distance. Based on the resulting dendrogram (Figure 2), it can be assumed that the optimal number of clusters is six.

Individual respondents (observations) were assigned to clusters using the k-variable method. This yielded six clusters of respondents with similar attitudes toward tea purchase and consumption. Descriptive statistics, in particular, median—Me and interquartile deviation—IQR (Table 5), were used to evaluate the characteristics of respondents in each cluster.

The analysis also used arithmetic means calculated for the new variables in the respondent groups (Figure 3). The averages were used only to better rank the results obtained.

Based on the results obtained, it is possible to distinguish homogeneous groups among the respondents taking part in the survey. These groups can be characterized as follows:

Cluster 1—“Occasional tea gourmets” n = 159 (9.4%) people who consume white tea far more often than others and red tea more often than others. These are almost exclusively women with the highest average education among all clusters,

Cluster 2—”Yerba mate drinkers” n = 159 (9.4%) people consuming yerba mate most often, also reaching for red and white tea more often than others. This group has the highest number of men, although women still make up the majority. Respondents in this group are also the youngest, have the best financial situation, are more likely to come from larger cities and have the worst education.

Cluster 3—“Tea gourmets” n = 180 (10.6%) people who consume red tea most often, but also frequently consume traditional varieties of tea (black, green, “herbal or fruit tea”), pay attention to brewing time and brew their tea rather short. This group includes more women than the average, they are more likely to come from smaller cities, and declare a slightly better financial situation and education.

Cluster 4—“Occasional consumers” n = 286 (16.8%) respondents in this group have been brewing tea longer than others but report lower frequency of consumption of teas and traditional teas. They are occasional consumers of stronger tea. This group consists of a larger than average number of men, rather slightly older, from smaller cities with the best financial situation and a low level of education.

Cluster 5—“Undemanding tea consumers” n = 766 (45.1%) people in this group declared the lowest values of all observed characteristics. They were the least likely to consume all types of tea, and the least likely to attach importance to all activities related to the purchase, preparation and consumption of tea. This group consists mainly of women, respondents in it are the oldest and in the worst financial situation.

Cluster 6—“Occasional strong tea consumers” n = 150 (8.8%) people who are the least likely to drink white tea and also the longest tea drinkers. This group includes a higher than-average number of men, they come from the smallest towns with the lowest education and almost the worst financial situation.

To determine whether demographic factors had a significant impact on tea consumption habits, tests were conducted to determine the significance of differences in average values across groups. Based on these, it can be concluded that men are more likely than women to consume white tea and traditional types of tea, take longer to brew them, and declare a greater attachment to the health benefits of tea. In the case of age, it can be concluded that younger respondents were more likely to consume white tea, yerba mate, pay more attention to the health properties of tea and generally consume more of it. Place of residence has an impact on yerba mate and white tea consumption. The larger the locality from which respondents come, the more likely they are to consume both types of tea.

Financial situation also differentiates respondents’ behavior toward tea. The better the situation, the less often the respondents treat tea as an everyday traditional beverage and at the same time have more knowledge about the health effects of tea. At the same time, those declaring that their financial situation is average pay the least attention to matters related to health, diet, etc.

The respondents’ education had an impact on the consumption of white and red tea. The higher the education, the more often these types of teas were consumed. At the same time, it could be observed that yerba mate was consumed more often in the middle-educated group and knowledge of the health properties of tea was greater, while tea was treated less often as an everyday drink.

## 4. Discussion

### 4.1. Type, Frequency, and Place of Tea Consumption

According to other authors [43,44,68] about 70–80% of Poles drink it every day, and 70–90% choose black tea. In this study, a significant percentage of respondents (over 80%) also indicated that they drink black and green tea, and a slightly smaller number (74.6–76.5%) indicated fruit and herbal infusions and aromatic tea. In the group of respondents, black tea was drunk several times a week, less often than reported by other authors [43,44]. The smallest number of respondents indicated oolong, white tea, and pu-erh teas. This was confirmed by other studies [68], according to which these teas are less known to Polish consumers and available in specialist stores.

According to other authors [75,76], the choice of teas depends on the age and gender of consumers. Red, white, and yellow teas were consumed occasionally. The elderly drank the most tea, which can also be explained by the power of motivation of older consumers who, in the face of emerging health problems, decide to use the health-promoting properties of tea or may result from attachment to the product.

Often the term tea is wrongly used to describe the assortment of plant materials and herbal mixtures used to make popular infusions, the so-called “fruit and herbal teas” [75].

This is due to the fact that such a trade name appears on the packaging of these beverages.

The respondents also indicated that they drink this type of infusion 1–3 times a month.

Yerba mate, which is not a tea, has recently appeared on the Polish market, was mentioned by 30% of the respondents but was characterized by a low frequency of consumption, which is confirmed by the results of other studies [75,76]. In previous studies, a lower percentage of Polish consumers surveyed indicated drinking green tea [43,68], similarly shown for British consumers (28%), which proves that English consumers are more attached to the centuries-old tradition of black tea consumption [68].

Teas are sold in many forms as a product to be infused in the form of leaves, granules, packed in bags (so-called express), or the instant type. Due to the wide assortment and convenience, express teas are gaining more and more supporters [69]. This trend is favored by factors such as an increase in the pace of life, change in production technology, quality improvement, and functional aspects. At the same time, shops specializing in selling teas have opened in the tea market in Poland [43]. A significant percentage of our respondents (over 60%) declared drinking express tea, which is confirmed by previous findings [44,67,77]. A smaller percentage of respondents choose teas with whole or crushed leaves and granulated. As indicated by other authors [77], leaf teas were used primarily by consumers of yellow tea and Oolong, and 38% of tea consumers used “herbal or fruit teas”.

On the other hand, even sophisticated British consumers, consuming green and white tea, bought it in sachets, preferring a comfortable lifestyle, at the expense of the sensory and health sensations that can be provided by tea brewed in the form of leaves [68].

The largest percentage of the respondents, regardless of their gender, consumed tea at home, rarely at work or with friends. It was unpopular to drink tea in a gastronomic establishment. Tea is usually associated with routine and ritualized household consumption. Other authors indicated similar places of tea consumption, with home first, work second, and bars and friend’s home third. According to these authors, in the northern part of Europe, tea was more frequently consumed at a friend’s place rather than at a bar [78].

The respondents consumed tea mainly between meals as well as during the first breakfast and dinner. Similar relationships were indicated by other authors, according to which Polish consumers have tea with the first breakfast (26%) and dinner (24%), but also with other meals and between meals (19%) [79]. According to Rusinek-Prystupa et al. [69], tea was most often drunk with meals (63.9% of responses), and almost half of the respondents drank tea on occasion or under various circumstances. Tea consumption was also indicated more often during visits than at work or at university (31.3% vs. 21.7%).

### 4.2. Factors Determining the Choice of Tea

Among the factors determining the choice of tea, the respondents mentioned quality, price, individual habits/preferences, brand, and flavor. Health reasons, promotion, packaging, and convenience were less important. The packaging size, friends’ opinions, and advertisements had a slight influence on their choice.

Similar results were obtained in another study [80], and the dominant factors were quality (38%), brand (22%), and low price (17%). Other authors also point to the significant role of the brand and the price of the packaged tea assortment in purchasing decisions [67,68]. For the customers of specialty tea stores, it did not matter, as they usually sell tea blends at a higher price, reflecting the quality, and in this case, the opinion of friends and the recommendation of store employees were more important. The special role of the price and quality of tea in shaping the behavior of tea consumers was noted by Li et al. [74]. The commercial attributes of products, such as the type, brand of tea, advertising impact, and packaging also play an important role in choosing tea [68].

The most popular brands among the respondents were Lipton, Dilmah, Tetley, Saga, Teekanne, and Twining’s. This is confirmed by the results of previous studies [69,71], according to which the most popular brands of tea were in the middle and low-price range, Lipton, and Saga. Some of the major commercial traders of black tea globally are Unilever-Lipton, PG Tips (Multinational); Associated British Foods-Twining’s (UK); Tata Tea-Tetley (India); Teekanne Group (Germany) [81], but in Poland, it is Unilever (Lipton, Saga), Herbapol Lublin, and Mokate. These three producers have nearly 70% of the market share (taking into consideration a small format of stores) in Poland [44]. Teleżyńska [82] stated that the most important sales channel was supermarkets, which accounted for over 38.4% of tea sales.

Among other factors of tea selection, the respondents indicated the sensory qualities of tea, such as taste and aroma, which is consistent with previous studies by other authors [68,70,71,75,76].

However, it was emphasized that these assessments were subjective, depending on the consumer’s individual sensory sensitivity. A type of tea was considered to be a more objective factor, which is conditioned by specific technological processes related to specific product characteristics [75,76]. Kozirok and Sitkiewicz [70] as important selection factors of tea, mentioned health-promoting properties, habituation, brewing convenience, price, friends’ opinion, and the brand of teas. Others, such as the origin of the product and promotional features (packaging, advertising), were not important for consumers.

### 4.3. The Methods of Preparing and the Kind of Tea Used by Respondents

Tea consumers have a wide variety of tea preferences. They prefer the traditional serving in a cup with a sachet or a large glass jug. However, they do not rule out other methods. New trends such as cold brew tea were not popular with our respondents. According to the SW Research [83] agency, 80% of surveyed Poles drink black tea every day or several times a day, starting their day with it or drinking it during meals or at social meetings, brewing in a teapot or a mug with a strainer.

The choice of the tea brewing time by respondents was caused by the different effects of the teas. Only one-third of respondents always pay attention to the method of brewing which affects the color, taste, and aroma of tea, as well as its strength, which is also related to the content of caffeine and L-theanine and depends on the type of tea [7,84,85,86,87].

More than half of the respondents chose the stimulating effect of black tea and a brewing time of up to 3 min, with another 25% of the respondents choosing a brewing time of up to 5 min together with the calming effect. The respondents used longer brewing times for white, green, and red tea. Dmowski et al. [68] indicated that Poles chose the health effects of teas and, to a lesser extent, the calming and stimulating effects. Most of them used up to 1.5 teaspoons of tea for one cup/glass, which is not a strong infusion.

Water used to brew tea was more important to the respondents, and a fairly high percentage (45%) used filtered water, Oligocene water, or mineral water for brewing tea.

The respondents chose a variety of tea additives, such as lemon and honey. Tea with lemon has been shown to have a beneficial effect on the body, reducing appetite, and honey has an antibacterial effect [88,89], however, the combination of hot tea with honey and lemon has no health properties and instead is one of the stereotypes passed down from generation to generation [90].

The respondents drink tea rather sporadically. According to other studies [91], average Pole consumes two–three cups of tea a day, but about 20% of the population consumes four to five teacups. In studies conducted in 27 European countries [78], the mean daily intake of tea in women and men ranged from 14–18 g/day (which corresponded to ~0.1 cups) in Spain to 788–928 g/day (~4.3 to 4.9 cups) in the UK general population. The lowest consumption of herbal tea was observed in Sweden (0 g and 7 g per day for men and women, respectively), and the highest in Germany (128 g/day for men and 202 g/day for women).

### 4.4. Knowledge about Tea by Respondents

In general, the respondents’ knowledge about the health properties of tea was not very high, and stereotypes were repeated, despite the fact that nearly 61% of the respondents assessed their nutritional knowledge as good and very good.

Contemporary, conscious consumers more and more often pay attention to the authenticity and health benefits of the product. They realize that teabags may contain fewer bioactive ingredients, positively influencing health, and one chooses their convenience [92,93].

### 4.5. Limitation

The strength of our study is the relatively large representative sample of adult Poles (n = 1700). We are aware that in the case of people who do not drink tea, our group was not a representative population for the entire adult population in our country, which may be a barrier to this study. However, it came from all over Poland, and its selection took into account all determinants also included in the group of tea consumers. Therefore, it can be considered as a reference group for comparative studies. Using an online survey to collect data can be considered a benefit as it allows the possibility of reaching a larger group of people from different backgrounds, which was very valuable when collecting data during a pandemic. On the other hand, it was also a limitation resulting from the possibility of participation only by people with internet access.

## 5. Conclusions

Based on the obtained results, we identified the habits and preferences of Poles related to tea drinking. The respondents preferred black tea, which they drink mainly at home, several times a week, during breakfast, and between meals. The respondents’ choice of tea was dependent on its quality, and above all, on its taste and aroma. The most popular brand among the respondents was Lipton. Most of the respondents did not pay attention to the method of brewing and poured boiling tap water over the tea bags and brewed them for up to 3 min. A significant percentage of the respondents drink teas without any additives.

Among consumers, depending on demographic characteristics, six clusters were identified differing in the type and frequency of tea consumption, being “Occasional tea gourmets”, “Yerba mate drinkers”, “Tea gourmets”, “Occasional consumers”, “Undemanding tea consumers”, and “Occasional strong tea consumers “. The most numerous (45%) group was the “Undemanding tea consumer”, while the remaining groups were similar in numbers.

Tea consumers do not have full knowledge about tea and the method of brewing. It can be concluded that the Polish tea consumer prefers conventional tea brewing methods, mainly reaching for black tea (a “traditionalist”) but is also open to novelties and the search for new flavors and types, reaching for less popular teas such as white or yerba mate. This is a niche that could be used in gastronomy to organize workshops on brewing various teas and educate consumers about the health-promoting properties of tea and the variety of its qualities. A good solution could also be to prepare a tasting menu based on a variety of teas.

Knowledge of consumer preferences regarding tea consumption, as well as determining the factors influencing their choice, can be an important source of information for producers and traders of these stimulants. Consumer segmentation can be helpful in conducting marketing activities to the appropriate group of consumers and can be of interest to other populations for cross-cultural comparison.

Differences in tea-drinking habits such as type of tea, the preparation processes, amount of tea consumed, and additions such as sugar, milk, and others may vary by population and country and could contribute to the inconsistencies found between studies comparing tea consumption.

## Figures and Tables

**Figure 1 foods-11-02873-f001:**
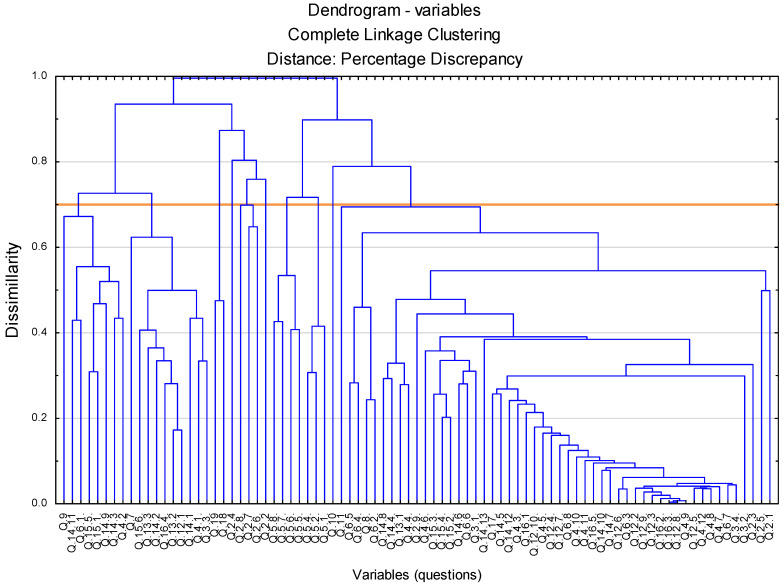
Dendrogram of variables agglomeration.

**Figure 2 foods-11-02873-f002:**
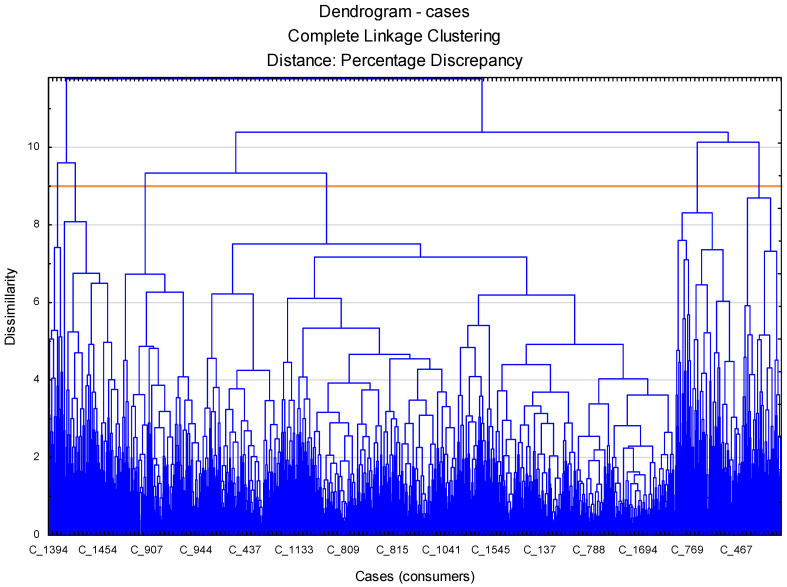
Dendrogram of consumers agglomeration.

**Figure 3 foods-11-02873-f003:**
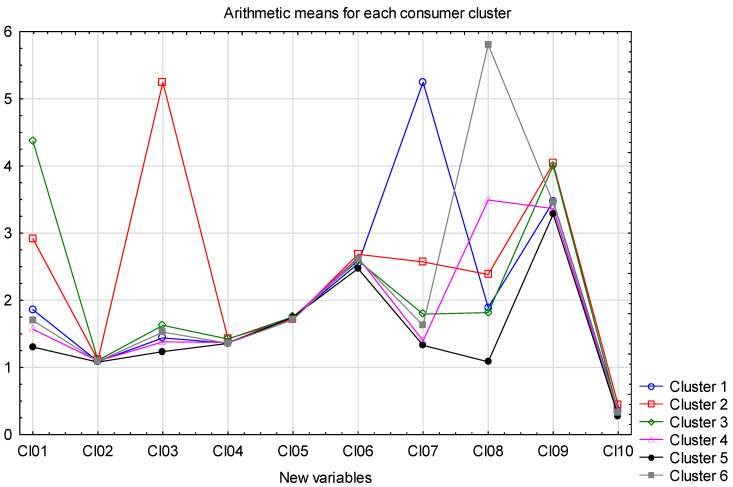
Arithmetic averages of new variables in the separated groups of respondents.

**Table 1 foods-11-02873-t001:** Characteristics of the surveyed sample of respondents.

Population Features	Group	Number of Respondents (n)	Percentage of Respondents (%)
Total	-	1700	100.0
Gender	women	1298	76.4
	men	402	23.6
Age	18–24 years old	731	43.0
	25–40 years old	611	35.9
	41–60 years old	285	16.8
	61–75 years old	73	4.3
Education	vocational or primary school	193	11.4
	secondary school	466	27.4
	higher education (university)	1041	61.2
Dwelling place	village	440	25.9
	city below 20,000 inhabitants	179	10.5
	city between 20,000–100,000 inhabitants	294	17.3
	city over 100,000 inhabitants	787	46.3
Financial status	below average (poor)	66	3.9
	average	754	44.3
	over average (good)	698	41.1
	very good	182	10.7

**Table 2 foods-11-02873-t002:** The type and frequency of tea drinking by respondents.

Tea	Respondents		Average * ± SD	Median *	Q25	Q75
	n	%				
White tea	720	42.4	1.9 ± 1.4	1	1	2
Green tea	1422	83.7	3.4 ± 1.8	3	2	5
Oolong tea	370	21.8	1.4 ± 0.9	1	1	1
Black tea	1512	88.9	4.4 ± 2.0	5	3	6
Red tea (pu-erh)	774	45.5	1.9 ± 1.3	1	1	2
Aromatic tea	1269	74.6	3.3 ± 1.9	3	1	5
Other, understood by respondents as tea						
Yerba mate	557	32.8	1.7 ± 1.4	1	1	2
“Fruit tea”	1301	76.5	3.1 ± 1.8	3	2	5
“Herbal tea”	1278	75.2	3.2 ± 1.9	3	1	5

* Scale: (1) never; (2) rarely than once a month; (3) 1 or 3 times a month; (4) once a week; (5) several times a week; (6) once a day; (7) several times a day.

**Table 3 foods-11-02873-t003:** Respondents preferences during the preparation of tea infusion.

Preferences	Respondents		Preferences	Respondents	
	n	%		n	%
Water used to brewing			Pay attention to way of brewing		
Tap water from water supply	931	54.8	Yes	547	32.18
Oligocene water	32	1.9	No	514	30.24
Mineral water no sparkling	53	3.1	Sometimes	639	37.59
Filtered water	684	40.2			
The amount of tea for 1 cup/glass (200 mL water)			Preferred brewing time		
Less than 1 teaspoon (<2 g)	318	18.7	Less than 3 min	478	28.1
1 teaspoon (2 g)	811	47.7	3 min	396	23.3
1.5 teaspoon (3 g)	338	19.9	4 min	210	12.4
2 teaspoon (4 g)	129	7.6	5 min	207	12.2
other	104	6.1	6 min	40	2.4
-	-	-	over 6 min	130	7.6
-	-	-	I don’t know, it isn’t important for me	239	14.1

**Table 4 foods-11-02873-t004:** Respondents’ opinion about tea.

Respondents’ Opinion/Knowledge	Average * ± SD	Median *
Drinking about 450 mL of tea a day reduces the risk of cardiovascular diseases	0.41 ± 0.70	0
Tea lowers blood pressure	0.23 ± 0.68	0
Drinking green tea reduces the risk of osteoporosis	0.00 ± 0.00	0
Tea reduces the risk of hyperlipidemia (high blood lipids)	0.23 ± 0.61	0
Strong tea drink before going to sleep makes it difficult to fall asleep	0.50 ± 1.10	1
The greatest amounts of components that have a beneficial effect on the human body are found in white and green tea	0.56 ± 0.91	0
Drinking too much tea can cause kidney stones	0.16 ± 0.86	0
Green tea reduces the appetite	0.13 ± 0.88	0

* Scale: (−2) definitely do not agree; (−1) do not agree; (0) neither agree or disagree—I don’t know; (1) agree; (2) definitely agree.

**Table 5 foods-11-02873-t005:** Values of medians (Me) and quartile ranges (IQR) in the separated groups of respondents.

Cluster	n		Cl01	Cl02	Cl03	Cl04	Cl05	Cl06	Cl07	Cl08	Cl09	Cl10	Gender	Age	Dwelling Place	Financial Situation	Education
1	159	Me	1	1.08	1	1.36	1.75	2.50	5	2	3.6	0.29	2	2	3	3	3
		IQR	1	0.05	1	0.14	0.17	0.75	2	2	1.8	0.43	0	1	2	1	1
2	159	Me	3	1.10	5	1.43	1.75	2.75	2	2	4.0	0.29	2	2	4	3	3
		IQR	3	0.10	2	0.14	0.17	0.75	3	3	1.6	0.57	1	1	2	1	1
3	180	Me	4	1.10	1	1.43	1.75	2.50	2	2	4.0	0.29	2	2	3	3	3
		IQR	2	0.08	1	0.29	0.17	0.75	1	1	1.6	0.57	0	1	3	1	1
4	286	Me	1	1.08	1	1.36	1.75	2.50	1	3	3.4	0.29	2	2	3	3	3
		IQR	1	0.08	1	0.14	0.25	0.75	1	1	1.4	0.57	1	1	3	1	1
5	766	Me	1	1.08	1	1.36	1.75	2.50	1	1	3.2	0.14	2	2	3	2	3
		IQR	1	0.05	0	0.14	0.17	0.75	1	1	1.4	0.57	0	1	3	1	1
6	150	Me	1	1.08	1	1.36	1.71	2.50	1	6	3.4	0.14	2	2	3	3	3
		IQR	1	0.08	1	0.14	0.25	0.75	1	0	1.4	0.71	1	1	3	1	1

Median—Me and interquartile deviation—IQR.

## Data Availability

The data presented in this article are available on reasonable request, from the corresponding author.

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
