# Peer review of "Consumer Choices and Habits Related to Tea Consumption by Poles"

_foods, 2022, doi:10.3390/foods11182873_

Round 1

Reviewer 1 Report

1-I'm not sure how accurate this statement is. (line 28)

Is this what was meant?

‘At the end of the study, tea drinking habit profiles of consumers in Poland were determined.’

2-The results of the study were not included in the summary.

3-It is not clear what is meant by the tea segment.

4-There is confusion at the entrance, skipped from tea to coffee, is only tea and coffee consumed as a beverage?

5-The health of tea was mentioned, then the marketing was started and health was said again, the entrance was messy.

6-The same clutter and confusion is present in the discussion section. I think there is a problem with logical flow in data transfer.

7-It is a good example that such a study can be done using the internet.

Author Response

Dear Reviewer,

We are excited to re-submit the improved and changed version of our manuscript titled: “Consumers' choices and habits related to tea consumption by Poles”.

We have addressed all issues indicated in the review report, and believe that the new version will meet the journal publication requirements. Thank you very much for the thorough review and insightful feedback. The text has been checked by  Smaller Earth Proofreaders. Please find our responses to You in the file attached. Thank you for your patience and help.

Yours sincerely,

Authors

Reviewer 2 Report

Relevance of the topic and study design:

This investigation is relevant, original and relatively well conceived in terms of questionnaire planning and data collection.

Title, abstract and keywords:

The title seems to fit the experimental work developed.

The abstract is globally adequate, but needs to be reformulated in view of major problems detected regarding the statistical analysis of the data.

The choice of keywords was appropriate

Introduction:

The introduction correctly helps to frame and contextualize the work. It presents the state of the art on a number of topics which are essential to the work that was carried out.

Materials and methods:

The description of the experiments conducted is fairly Ok. I have some observations, however.

In line 144: The Statsoft corporation is not a Polish enterprise, and the headquarters of the company are not in Krakow-Poland. Please correct by putting the right city/Country, which is Hamburg – Germany for the European Branch.

Line 163: With your data you can’t use parametric tests and therefore you can’t use ANOVA, which is a parametric test.  You must use Kruskal-Wallis, which you probably dis and even mentioned it but somehow included by mistake the mention to ANOVA.

Table 2 and the rest of the paper: If there are situations where the use of word tea dos not apply, then you should correct in all times this occurs. Tea is the word used conceptually exclusively for infusions of the plant Camellia sinensis. So in all other situations the beverages are not tea but infusions. I understand that people commonly mistake it by tea and name it tea. So my suggestion is to use the word ‘tea’ but with quotes as an indication of being a different usage of the word not corresponding to its correct meaning.

So you use Black tea or Green tea but use fruit ‘tea’.

With respect to the treatment of the date I find major issues relating to the application of statistical techniques.

The recommended analysis to group variables should be factor analysis not cluster analysis. Cluster analysis is the recommended technique for grouping the cases (in this case RESPONDENTS) NOT THE VARIABLES.

So, even though the data collection was OK, the data treatment has major problems, and either it is completely reformulated or the work must not be published.

Results and discussion:

The results presented that concern only descriptive statistics are in general fine.

Author Response

(The authors gave the same response as above.)

Reviewer 3 Report

This article "Consumer choices and habits related to tea consumption by Poles” was major revised and has not a novelty.

Title: It is perfect.

Abstract:

·         Line 18: purchasing consumer behavior? Or consumer purchasing behavior?

·         Using cluster analysis, we identified six groups of tea consumers. There are “Occasional tea gourmets”, “Yerba mate drinkers”, “Tea gourmets”, “Occasional consumers”, “Undemanding tea consumers,”, “Occasional strong tea consumers” what is the meaning of the gourmet or yerba? Please explain them in introduction.

·         Based on study results possible to know the tea drinking habits in Poland. Why did not  you do in your currently study?

·         Design and optimization of the process were provided by statistical software Minitab 17, through Plackett-Burman and Central Composite Design. Please consider the number of the axial, central and factorial points in design.

·         Why did you use bovine colostrum?

·         Please point the optimal conditions which you achieved in the Abstract?

·         However, the samples to which oleogel was added instead of milk fat lacked butyric and caproic acids, both of which are aromatic fatty acids. Aromatic acid or aliphatic acid? Please consider to it.

Keywords: Please choose keywords other than the main words of the title. In this case, other researchers can find your article by searching a wide range of words through databases. I propose another keywords as the follow:

consumer habits, tea, Tea brewing, Consumer preferences, Poles, Computer assisted web interviewing,

Introduction:

·         The purpose of the research should be stated in a scientific manner and by stating the treatments used well and in detail. Do not use question sentences in this section.

·         In the last paragraph of the introduction, please state the conditions of the operation as well as the treatments.

Materials:

Methodology:

·         Why did not you select nearly equal for “Percentage of respondents (%)”?

 “Results:

·         Table 2: Average ± SD What does it mean? What is the meaning of the Median Me? Are you right Me?

·         Table 4: It seems that the average numbers should be moved with the standard deviation and what id the rol of Median in this table? Or research.

·         Table 2:*Scale: (-2) Definitely do not agree; (-1) Do not agree; (0) neither agree or disagree – I don’t know; (1) Agree; (2) Definitely agree. These numbers do not have an example in the table

·         Figure 1: please consider to it that each table and figures must to self-explanatory. 

Discussion:

Discussion text must grammar improve and in some cases it is very weak and maybe there is no discussion at all.

Conclusions:

Conclusion is very general, try to make it more scientific, comprehensive and concise in detail, especially.

References: It is OK.

·         The article has many flaws in express and concept of English, it is suggested to be revised in a scientific and native way.

·         I don't know what the results of this research really solve in the world of problems related to tea and it seems that the results are regional and it is better to be published in the same country as Poland.

Author Response

(The authors gave the same response as above.)

Round 2

Reviewer 2 Report

The revised version can be acpted for publication.

Author Response

Dear Reviewer,

Thank You very much for your comments and for accepting our revision.

Authors